# Acute Effects of Butyrate on Induced Hyperpermeability and Tight Junction Protein Expression in Human Colonic Tissues

**DOI:** 10.3390/biom10050766

**Published:** 2020-05-14

**Authors:** Mathias W. Tabat, Tatiana M. Marques, Malin Markgren, Liza Löfvendahl, Robert J. Brummer, Rebecca Wall

**Affiliations:** Nutrition-Gut-Brain Interactions Research Centre, Faculty of Health and Medicine, School of Medical Sciences, Örebro University, 701 82 Örebro, Sweden; mathias.tabat@oru.se (M.W.T.); tatiana.marques@oru.se (T.M.M.); maaalinm@live.se (M.M.); liza.lofvendahl@oru.se (L.L.); robert.brummer@oru.se (R.J.B.)

**Keywords:** intestinal barrier function, butyrate, intestinal permeability, tight junctions, Ussing chamber

## Abstract

Intact intestinal barrier function is essential for maintaining intestinal homeostasis. A dysfunctional intestinal barrier can lead to local and systemic inflammation through translocation of luminal antigens and has been associated with a range of health disorders. Butyrate, a short-chain fatty acid derived from microbial fermentation of dietary fibers in the colon, has been described as an intestinal barrier-strengthening agent, although mainly by using in vitro and animal models. This study aimed to investigate butyrate’s ability to prevent intestinal hyperpermeability, induced by the mast cell degranulator Compound 48/80 (C48/80), in human colonic tissues. Colonic biopsies were collected from 16 healthy subjects and intestinal permeability was assessed by Ussing chamber experiments. Furthermore, the expression levels of tight junction-related proteins were determined by quantitative reverse transcription polymerase chain reaction (qRT-PCR). Pre-treatment with 5 mM butyrate or 25 mM butyrate did not protect the colonic tissue against induced paracellular or transcellular hyperpermeability, measured by FITC-dextran and horseradish peroxidase passage, respectively. Biopsies treated with 25 mM butyrate prior to stimulation with C48/80 showed a reduced expression of claudin 1. In conclusion, this translational ex vivo study did not demonstrate an acute protective effect of butyrate against a chemical insult to the intestinal barrier in healthy humans.

## 1. Introduction

The intestine is the main organ involved in the uptake of nutrients and water. The intestinal barrier provides an essential separation between the intestinal lumen and the internal body environments and, thereby, between luminal antigens and the body’s immune system. As the subject of constant dynamic regulation, intestinal barrier function helps the human body to react to external influences and to maintain homeostasis. The intestinal barrier consists of a physical barrier (intestinal mucus layer, epithelium and underlying tissue including the vascular endothelium) and a chemical barrier (digestive secretions, anti-microbial peptides, immune molecules and inflammatory mediators) [1]. A well-functioning intestinal barrier, thereby, allows the uptake of nutrients and simultaneously serves as an appropriate defense against the translocation of potentially harmful substances such as lipopolysaccharides (LPSs) or microorganisms that could trigger the immune system. The passage of these substances through a dysfunctional intestinal barrier may lead to local or systemic inflammation and thus have negative consequences for both intestinal and systemic health. Ongoing inflammation processes can, in turn, result in a compromised intestinal barrier, leading to a vicious circle. Accordingly, intraperitoneal injections of clinically relevant doses of LPSs in mice lead to a pro-inflammatory state in the intestine and increased intestinal tight junction permeability [2]. In humans, altered intestinal permeability has been associated with the pathogenesis of several gastrointestinal disorders including inflammatory bowel diseases (IBDs), irritable bowel syndrome (IBS) and celiac disease [3,4]. Beyond the gut, increased intestinal permeability has been associated with diseases such as diabetes [5] and autism [6,7]. Further, these diseases have also been associated with low-grade systemic inflammation, gut microbiota dysbiosis and lower levels of butyrate-producing bacteria in the gut [8,9,10,11,12,13].

Butyrate is a short-chain fatty acid (SCFA) derived from microbial fermentation of dietary fibers in the colon. Luminally produced butyrate is rapidly absorbed by the colonic mucosa and almost completely used as an energy source by colonocytes [14,15,16]. Only relatively small amounts of butyrate enter the bloodstream [17,18,19]. As butyrate is known to be a histone deacetylase inhibitor with, among others, reported anti-inflammatory, oxidative stress-reducing and intestinal barrier strengthening effects [20,21], it has become an interesting research target for novel treatment and prevention strategies.

Several studies, using cell culture models and animal models, have shown that butyrate can strengthen barrier function and decrease intestinal permeability [22,23,24,25,26,27]. Paracellular permeability is, to a great extent, controlled by tight junctions, and disrupting their integrity and assembly results in increased permeability (“leaky gut”) [28]. Butyrate has been shown to affect the expression of tight junction proteins including claudin 1, claudin 2, claudin 7 and occludin in human cell culture models and in the small intestine of rats [29,30,31,32,33,34], suggesting this as a possible mechanism by which butyrate beneficially affects intestinal permeability. However, most of the knowledge on butyrate’s effect on barrier function and intestinal permeability comes from in vitro work and animal models and more studies addressing butyrate’s role on intestinal permeability in humans need to be performed before we can consider its therapeutic use. Thus, the aim of this study was to investigate butyrate’s ability to prevent induced intestinal hyperpermeability in human colonic tissues by applying an ex vivo setting.

## 2. Materials and Methods

### 2.1. Study Participants and Ethics

Sixteen healthy subjects aged between 18 and 65 years were included in the study. Subjects were recruited via advertisements placed at Örebro University campus. Exclusion criteria included previous complex abdominal surgery, a hypertonic condition demanding medical treatment, a diagnosed psychiatric disease, usage of prescribed medications (except oral contraceptives) during 14 days preceding the intervention, premenstrual syndrome, being pregnant or breast feeding, or a diagnosed gastrointestinal disease (e.g., IBDs). The study was approved by the Regional Ethical Review Board in Uppsala (Dnr 2013/037). The principles of the Helsinki declaration were followed throughout the study and all participants signed an informed consent before participation.

### 2.2. Ussing Chamber Experiment

#### 2.2.1. Collection of Colonic Biopsies

Study participants (n = 16) underwent a sigmoidoscopy in the morning after 10 h fasting. No bowel cleansing procedure was performed in order to avoid affecting mucosal integrity. Colonic biopsies were obtained from the unprepared sigmoid colon, at the crossing with the common ileac artery, using non-spiked Captura biopsy forceps (Cook Medical, Bloomington, IN, USA) and were immediately transferred to ice-cold oxygenated modified Krebs–Ringer bicarbonate buffer (aqueous solution with 115 mM NaCl, 1.25 mM CaCl_2_, 1.2 mM MgCl_2_, 2 mM KH_2_PO_4_ and 25 mM NaHCO_3_, set to a pH of 7.2 with 1 M hydrochloric acid solution and then oxygenated with gas containing 95% O_2_ and 5% CO_2_; from now on, called KRB). Biopsies were transported in KRB buffer to the laboratory within 10 min.

#### 2.2.2. Experimental Setup

A total of twelve biopsies from each participant were used per experiment. Six biopsies were used as experimental controls—three biopsies were left unstimulated and three biopsies were stimulated with 10 ng/mL C48/80 (Sigma-Aldrich, Saint Louis, MO, USA), a previously described mast cell degranulator used to induce hyperpermeability [35,36]. The remaining biopsies were divided in two treatment groups—three biopsies were pre-treated with 5 mM sodium butyrate (Sigma-Aldrich) and three biopsies were pre-treated with 25 mM sodium butyrate before stimulation with 10 ng/mL C48/80.

#### 2.2.3. Experimental Procedure

Biopsies were mounted in 1.5 mL Ussing chambers (Harvard apparatus Inc., Holliston, MA, USA). Within the chambers, the biopsies were held in between two polyester films that exposed a round area of 1.77 mm^2^ of the mounted biopsy, as described previously [37]. Both half chambers were filled with ice-cold KRB. Buffers facing the serosal side of the tissue contained glucose (0.01 M), whereas buffers on the mucosal side contained mannitol (0.01 M). Throughout the experiment, every chamber was continuously oxygenated with 95% O_2_ and 5% CO_2_ and held at 37 °C. To monitor tissue viability, the electrophysiological parameters transepithelial electrical resistance (TER), potential difference (PD) and short-circuit current (Isc) were measured every 30 s throughout the experiment. Biopsies with a PD > 0.5 were excluded due to uncertain viability [37]. After 10 min of equilibration, buffers on both sides were exchanged with fresh 37 °C warm buffers. Sodium butyrate (Sigma-Aldrich) was added to the mucosal side of the chamber, with a final concentration of 5 or 25 mM. Twenty minutes thereafter, C48/80 (10 ng/mL) or KRB (unstimulated group) was added to the serosal side of the designated chambers. The C48/80 concentration used has previously been optimized for Ussing chambers within our laboratory [38]. The paracellular permeability marker FITC-dextran 3000–5000 (FD4, Sigma-Aldrich; 2.5 nM) and the transcellular permeability marker 45 kDa protein horseradish peroxidase (HRP; Type IV, Sigma-Aldrich; 5.38 µM) were added to the mucosal side. Permeability was assessed at baseline (T0), 30 min (T30) and 60 min (T60) after the markers were added. After the Ussing experiment, biopsies were quickly removed from the chambers and stored in RNALater (ThermoFisher Scientific, Waltham, MA, USA) at 4 °C overnight and further kept at −20 °C until RNA isolation.

#### 2.2.4. Measurement of Median TER

Baseline TER values were measured in a time window of 10 min before any treatment or stimulation (time period between T-30 and T-20), and median TER values (MTER) were calculated. MTER was used to access baseline paracellular integrity and to exclude any significant selection bias before treatments were applied.

#### 2.2.5. Measurement of FITC-dextran and HRP

FITC-dextran passage was determined by fluorescence measurement at Δ_ex_ = 485 nm and Δ_em_ = 530 nm using an EnSpire Multimode Plate Reader (PerkinElmer, Waltham, MA, USA). Horseradish peroxidase was measured by ELISA using the QuantaBlu Fluorogenic Peroxidase Substrate Kit (Thermo Fisher Scientific) as previously described [38]. Measurements were performed in duplicates with a standard curve. FITC-dextran and HRP passages are expressed as ΔT60-T0.

### 2.3. RNA Isolation

Biopsies were homogenized with a TissueRuptor (Qiagen, Venlo, The Netherlands) in 500 µL Trizol reagent (Thermo Fisher Scientific) per biopsy. The manufacturer’s instructions were followed until phase separation with the addition of two changes: 200 µL chloroform was used per biopsy and the centrifugation was performed at 21,000× *g*. The aqueous phase was then transferred to a Qiagen RNeasy mini column and the manufacturer’s instructions were followed to isolate RNA. RNA was eluted in diethylpyrocarbonate (DEPC)-treated water (Invitrogen by Thermo Fisher scientific, Waltham, MA, USA) and RNA quantity and quality were assessed by a BioAnalyzer 2100 using a RNA Nano 6000 kit (Agilent technologies, Santa Clara, CA, USA), according to the manufacturer’s instructions. RNA integrity numbers (RIN) of 5 and higher were accepted for qRT-PCR analyses. All the biopsies from one participant were excluded because of poor RNA quality.

### 2.4. Quantitative Reverse Transcription Polymerase Chain Reaction (qRT-PCR)

Complementary DNA (cDNA) of 100 ng isolated RNA transcripts were made using a Superscript Vilo cDNA synthesis kit (Thermo Fisher Scientific) with a 20 µL final volume according to the manufacturer’s instructions, and subsequently stored at −20 °C.

For the real-time PCR, 1.5 µL cDNA in 13.5 µL mastermix was used per well in a 96-well MicroAmp fast optical reaction plate (Thermo Fisher Scientific). The mastermix contained 7.5 µL LuminoCt qPCR Ready Mix (Sigma-Aldrich), 0.45 µL ROX dye (Sigma-Aldrich), 0.375 µL TaqMan FAM-MGB probe (Thermo Fisher Scientific) and 5.175 µL PCR-grade water (Sigma-Aldrich) per well. The TaqMan probes used were claudin 1 (Hs00221623_m1CLDN1), claudin 2 (HS00252666_s1CLDN2), claudin 7 (HS00600772_m1CLDN7), occludin (HS00170162_m1OCLN), and IkB-alpha (Hs00355671_g1NFKBIA). *GAPDH* (probe: Hs99999905_m1GAPDH) was used as a reference gene. The qPCR reaction was performed in a 7900HT Fast Real-Time PCR System (Applied Biosystems by Thermo Fisher Scientific, Waltham, MA, USA) with one cycle at 95 °C for 20 s, and 40 subsequent cycles at 95 °C for 2 sec and 60 °C for 20 sec. A fixed threshold among plates was used per probe. Triplicate values of Ct were accepted when the standard deviation was below 0.3 and a PCR-efficiency of 100 ± 10 % was verified by a standard curve of serial dilutions. Relative quantification was performed by calculating the ΔΔCt according to Livak and Schmittgen [39] using Ct_reference gene_-Ct_gene of interest_ as the ΔCt. The qPCR results are presented as log fold changes (ΔΔCt) of unstimulated or C48/80 plus butyrate-treated vs. C48/80–stimulated biopsies alone.

### 2.5. Statistical Analyses

The significance of treatments in FITC-dextran and HRP passage in Ussing experiments were analyzed by repeated measures one-way ANOVA and Bonferroni’s post-hoc multiple comparisons correction. Differences in MTER and the reduction in relative TER were tested by one-way ANOVA with Bonferroni post-hoc multiple comparison correction and the Kruskal–Wallis test with Dunn’s post-hoc multiple comparison test, respectively. Gene expression data was analyzed on ΔΔCt values using the Friedmann test with Dunn’s post-hoc multiple comparisons test. Correlations between age and intestinal permeability were tested using linear regression. The influence of sex on the effects of the stressor C48/80 or butyrate on permeability was tested with a two-way ANOVA and Sidak’s post-hoc multiple comparisons correction. Normal distribution was tested with the Shapiro–Wilk test. For normally distributed data, ANOVA tests were used. For non-parametric data, the Kruskal–Wallis test and the Friedmann test were used. Differences were considered significant at *p* < 0.05 and trends for statistically significant differences were recognized at *p* < 0.10.

## 3. Results

A total of 16 experiments were conducted. A threshold of 20% difference between the unstimulated group and the C48/80 group in either of the permeability markers (FITC-dextran or HRP) was established in order to assure a substantial stressor effect. Five experiments were excluded because the stressor C48/80 failed to increase intestinal permeability in the biopsies tested and one experiment was excluded because of technical problems with the Ussing chambers. Only results from the experiments with increased permeability are included in the results and discussion if not stated differently. The full dataset is visualized in Appendix A of the supplements. Information about the distribution of age and sex in the study population is shown in Appendix A.

### 3.1. Electrophysiological Changes

All groups had similar paracellular integrity before treatments and stimulations, as shown by the not significantly different median transepithelial electrical resistance (MTER) values in Figure 1.

The electrophysiological parameters TER, PD and Isc were monitored over time. PD was confirmed to be within a viable range throughout the experiments (≤ 0.5 mV). Moreover, the absolute TER values dropped similarly among the groups without any significant differences throughout the 60 min of the Ussing experiment (Appendix A).

Changes in relative TER values for the duration of the experiment are shown in Table 1, with 30 min intervals. TER at the beginning of the experiment (T0) was set as 100%, and values for T30 and T60 were normalized to T0. No significant differences were observed between groups, but a tendency (*p* = 0.062) towards lower relative TER was seen after 60 min in the C48/80 plus 25 mM butyrate group compared to the C48/80 group.

### 3.2. Pre-Treatment with Butyrate Did Not Affect Human Intestinal Permeability Ex Vivo

Colonic biopsies stimulated with C48/80 exhibited increased permeability, as demonstrated by a significantly higher passage of the paracellular marker FITC-dextran (*p* = 0.018) and the transcellular permeability marker HRP (*p* = 0.013) compared to the unstimulated biopsies (Figure 2). Pre–treatment with 5 mM butyrate or 25 mM butyrate did not significantly affect paracellular or transcellular permeability compared to biopsies that were stimulated with C48/80 alone (Figure 2). 

### 3.3. Butyrate Did Not Increase the Expression of Tight Junction Proteins in Colonic Biopsies

The expression level of claudin 1 was significantly decreased in the biopsies treated with C48/80 plus 25 mM butyrate compared to the biopsies stimulated with C48/80 alone (*p* = 0.002) (Figure 3). No other significant differences in the expression levels of claudin 1, claudin 2, claudin 7, occludin and NF–κB inhibitor alpha (IkB–alpha) were observed, both when comparing the C48/80 plus butyrate (5 and 25 mM) groups to the C48/80 group and when comparing the C48/80 group to the unstimulated group.

## 4. Discussion

In recent years, several studies have linked intestinal barrier dysfunction to a variety of disorders and diseases. Indeed, different diseases including IBDs, IBS, celiac disease and type 2 diabetes have been associated with increased intestinal permeability and are all characterized by sustained increased immune and inflammatory activity [3,4,40]. Substances such as LPSs that can evoke an immune response are abundant in the intestinal lumen. An intact intestinal barrier efficiently prevents translocation of these substances to the internal body environment. However, in a scenario of impaired barrier function with increased intestinal permeability, both local and systemic inflammation cascades might be triggered by the translocation of luminal components into the host. Impaired intestinal barrier function is now seen as a hallmark of IBDs, and immunity and intestinal barrier function appear to be closely interconnected to their pathophysiology on both the cellular and molecular levels [41]. Given the importance of the intestinal barrier for health, knowledge on how intestinal barrier function can be improved is of great value.

Butyrate has been described as an intestinal barrier-strengthening agent for more than a decade, mainly by using in vitro and animal models [42]. The results from the present study showed that during acute conditions, butyrate at physiological concentrations of 5 and 25 mM did not protect human colonic tissue against C48/80–induced hyperpermeability ex vivo. Although the biopsies incubated with C48/80 demonstrated a significant increase in both para- and transcellular permeability compared to the unstimulated biopsies, no differences were observed between the C48/80–stimulated biopsies and the biopsies that were pre-treated with butyrate prior to the stressor-stimulation. While the results from the present study demonstrate that butyrate does not exert acute regulatory effects on the intestinal permeability of human tissue ex vivo, butyrate may still have long-term regulatory effects. Several cell culture studies have shown that butyrate has the ability to increase the monolayer’s TER, a broadly accepted measurement reflecting paracellular permeability and tight junction integrity [43]. Indeed, butyrate concentrations between 1 and 5 mM were shown to significantly increase TER after 24 to 96 h in Caco-2 or HT-29 cell monolayers accompanied with a decreased paracellular permeability measured by inulin, mannitol or FITC-dextran passage [22,23,24,25,34,44]. In addition, a recent in vitro study showed that sodium butyrate, at concentrations ranging between 1 and 10 mM and 5 days incubation time significantly improved the epithelial barrier function in E12 human colon cells measured by TER and FITC-dextran passage, whereas higher concentrations (50–100 mM) showed no beneficial effects [45]. Butyrate has also been shown to partly counteract earlier induced damage to rat colonic tissue [27,46]. However, all the aforementioned studies were conducted using in vitro or animal models and, moreover, some preclinical studies have revealed contradictory findings to butyrate’s intestinal barrier-strengthening effects [47,48,49]. In a recent ex vivo study using monolayers of primary epithelial cells from subjects with and without ulcerative colitis, 48 h co-incubation with 8 mM butyrate increased TNFα and IFNγ-induced barrier impairment and enhanced inflammatory responses, although the gene expression levels of tight junction proteins indicated a strengthened barrier [49]. Since human studies assessing butyrate’s effects on intestinal barrier function and intestinal permeability are scarce, we employed an ex vivo human setting to explore the potential of an acute stimulation with butyrate to prevent stress-induced hyperpermeability in human colonic biopsies. By pre-treating biopsies with butyrate prior to adding a stressor (C48/80) that induces hyperpermeability, we were able to mimic a more biologically relevant preventive setting. In contrast to the present study, the previously mentioned studies have all determined the effect of butyrate without applying an insult to the model or tested whether butyrate rescues intestinal permeability by incubating butyrate after an induced stress.

C48/80 is known to induce mast cell degranulation, leading to the release of histamine, proteases and cytokines that, in turn, can increase tissue permeability. This has previously been shown in vitro with the release of tryptase affecting claudin 1-expression [50] and the assembly of claudin 1, occludin and zonula occludens 1 (ZO–1) in tight junctions [51]. Moreover, the release of TNF–alpha subsequently activated NF–κB and changed the protein levels of ZO-1 [52] and increased the expression levels of the pore-forming claudin 2 [53]. In the present study, stimulation with C48/80 did not significantly change the expression level of the tight junction proteins claudin 1, claudin 2, claudin 7 and occludin in the colonic biopsies. This could be due to insufficient incubation time or insufficient mast cell degranulation. To which extent mast cells underwent degranulation in the biopsies used in the present study was, however, not evaluated. Furthermore, it is possible that C48/80 affects tight junctions by altering or disrupting their assembly, changing internalization rates or the degradation of tight junction proteins [54], rather than affecting their expression. This could explain why we observed increased intestinal permeability in the C48/80 group compared to the unstimulated biopsies, but no differences in the gene expression levels of the assessed tight junction proteins. Additionally, other tight junction proteins that were not analyzed in the present study, such as ZO–1 or other claudins, could have been affected. The precise molecular mechanism of how C48/80 increases intestinal permeability, if not mediated through the expression of the tight junction proteins addressed in this study, remains to be further evaluated. Our results demonstrated a significantly lower expression of claudin 1, a sealing tight junction protein, in the biopsies that were treated with 25 mM butyrate compared to C48/80-stimulated biopsies alone. A reduced claudin 1 expression has been shown to increase paracellular permeability in several studies [31,55]. However, in the present study, we did not observe an increase in permeability in the biopsies treated with 25 mM butyrate prior to C48/80 compared to biopsies stimulated with C48/80 alone. Moreover, previous findings of cell culture studies, showing altered expression rates of claudin 1, claudin 2, claudin 7 and occludin following butyrate exposure [29,30,31,32,33,34,49], were not observed in the present ex vivo study. Further, a greater variation between replicates is observed using tissues compared to cell cultures [56].

Age and sex are possible factors influencing intestinal barrier function [57,58] and mast cell degranulation [59]. In our dataset, however, we could not find any significant correlations between age and measured permeability of unstimulated biopsies, nor between age and the effects of the stressor C48/80 or of butyrate on intestinal permeability (Appendix A). Furthermore, no significant differences were observed when comparing the effects of the stressor or butyrate on intestinal permeability between male and female participants (Appendix A). However, this has to be considered with caution, as the number of individuals of each sex is quite small.

A previous study by Vanhoutvin et al. detected an upregulation of IkB-alpha in healthy subjects after self-administration of 100 mM butyrate enemas daily for two weeks compared to placebo [60]. The authors suggested a potential anti-inflammatory effect of butyrate by the inhibition of NF-kB activation—an inhibition that could potentially also protect tight junctions and the paracellular barrier as shown in several other studies [52,61]. However, an altered expression of IkB-alpha was not observed in the present study. This might be due to the differences in the study design, the higher butyrate concentration used and/or the much longer exposure time to butyrate in the study by Vanhoutvin et al. [60].

Minor and non-significant differences in MTER including slightly higher TER values in the unstimulated group and slightly lower TER values in the C48/80 plus 25 mM butyrate group compared to the two other groups were observed in the present study. MTER is a baseline value before any treatment or stimulation has been performed, which could indicate a possible selection bias that was introduced during the mounting of the biopsies. As the absolute reduction in TER throughout the experimental duration was similar among the different treatment groups, the described variations in MTER could have led to the trend towards a bigger relative reduction in TER seen in the C48/80 plus 25 mM butyrate group. It is also possible that the higher concentration of 25 mM butyrate led to a higher decrease in TER. This corresponds well with the significantly lower claudin-1 expression in the 25 mM butyrate plus C48/80 group. However, this effect was not reflected in increased paracellular permeability in that group of biopsies.

Although preclinical studies generate important data, clinical translation remains a challenge. Discrepancies between our results and that of studies applying cell cultures and animal models may be related to different butyrate concentrations, to the limited incubation time during the Ussing experiment and to different mechanisms involved in the effects of butyrate on intestinal barrier function that were not covered in this study, such as secreted mucins, trefoil factors, transglutaminase, heat shock proteins or inflammatory factors. In the present study, we selected butyrate concentrations that resemble physiological conditions in the human distal colon. SCFA concentrations in fecal matter from the human proximal colon range from 70 to 140 mM and decline to 20 to 70 mM in the distal colon, with a general ratio of 3:1:1 for the three SCFAs, acetate, propionate and butyrate [17,62]. In stool samples, butyrate concentrations between 1.8 and 48.5 mmol/kg [63,64,65,66] and even up to 231.4 mmol/kg [67] have been reported. Hence, the butyrate concentrations used in the present study, 5 and 25 mM, are concentrations within a physiological range in the distal colon. In addition, the recommended concentration of butyrate used in in vitro models is currently 0–8 mM [31].

C48/80 can increase intestinal permeability by degranulating mast cells, thereby resembling a Corticotropin-releasing hormone (CRH)-mediated neuroimmune response to psychological stress which is believed to be a potential pathophysiological mechanism in gastrointestinal disorders connected to barrier defects [68]. It therefore seemed to be a valid stressor to be used in our experimental design. However, in agreement with previous observations in human colonic biopsies mounted in Ussing chambers [36], C48/80 did only significantly increase the permeability in 66% of the subjects included in this study, reducing the total number of experimental replicates.

## 5. Conclusions

Butyrate has previously been shown to strengthen the intestinal barrier and thus prevent the translocation of luminal antigens such as LPSs and subsequent inflammatory processes. This translational ex vivo study showed that an acute stimulation with butyrate, at physiological concentrations, did not protect human colonic tissue against increased permeability induced by a chemical stressor. Hence, further clinical studies are needed to clearly delineate the effects of butyrate on intestinal permeability and barrier function.

## Figures and Tables

**Figure 1 biomolecules-10-00766-f001:**
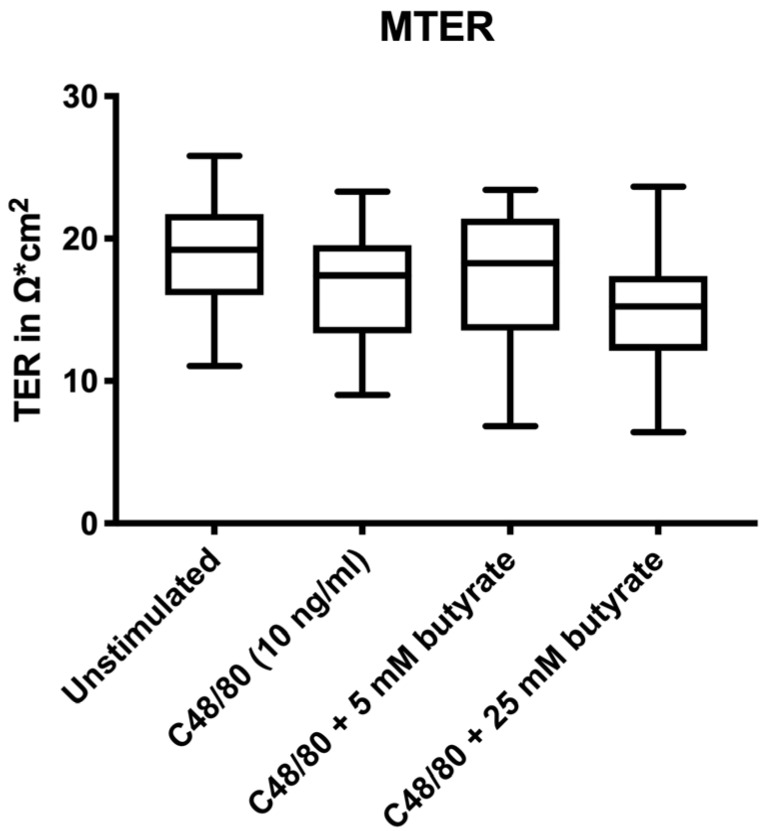
Median baseline TER (MTER) for the different treatment groups. Boxplots show MTER with the marked median, and whiskers visualize minimum and maximum values. MTER was determined within a time period of 10 min (T-30 to T-20) prior to any treatment or stimulation of colonic biopsies mounted in Ussing chambers. *n* = 10.

**Figure 2 biomolecules-10-00766-f002:**
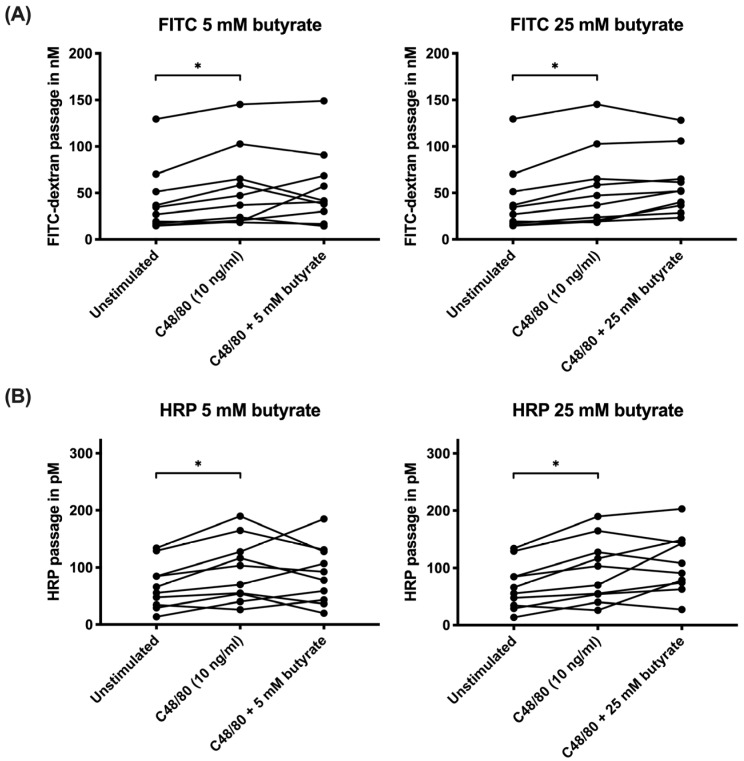
Effects of butyrate on intestinal permeability in colonic biopsies mounted in Ussing chambers. Paracellular permeability (**A**) and transcellular permeability (**B**) are displayed with dots connected by a line for each participant. Biopsies were analyzed in biological triplicates with no stimulation, stimulation with C48/80 (10 ng/mL) alone or in combination with 5 mM sodium butyrate or 25 mM sodium butyrate, respectively. * *p* < 0.05, *n* = 10.

**Figure 3 biomolecules-10-00766-f003:**
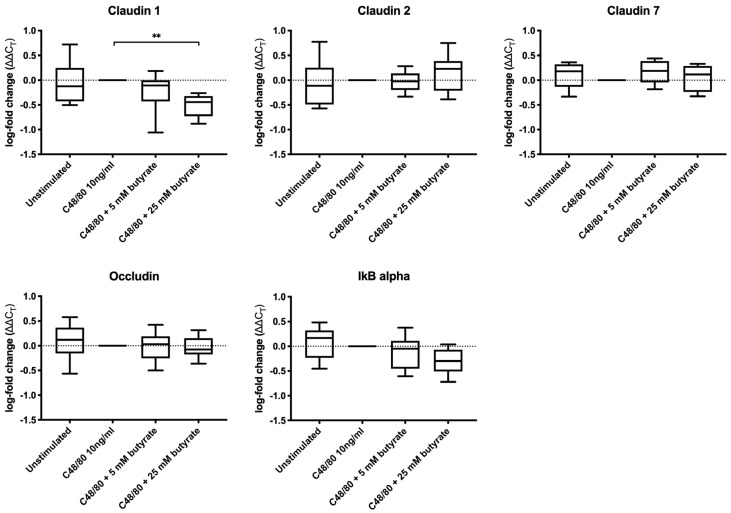
Gene expression levels of tight junction proteins and NF–κB inhibitor alpha (IkB–alpha). Boxplots show logarithmic fold changes of unstimulated or C48/80 plus butyrate-treated vs. C48/80–stimulated biopsies alone. Median values are indicated, and whiskers visualize minimum and maximum values. ** *p* < 0.01, *n* = 9.

**Table 1 biomolecules-10-00766-t001:** Relative transepithelial electrical resistance (TER) values (mean ± s.d.) normalized to baseline (T0).

Stimulation and Treatment	T0	T30	T60
Unstimulated	100.00	84.37 ± 9.75	80.43 ± 10.42
C48/80	100.00	88.49 ± 14.30	84.74 ± 16.25
C48/80 plus 5 mM butyrate	100.00	82.06 ± 9.05	78.29 ± 10.06
C48/80 plus 25 mM butyrate	100.00	76.68 ± 9.37	72.96 ± 9.48

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
