# Peer review of "Acute Effects of Butyrate on Induced Hyperpermeability and Tight Junction Protein Expression in Human Colonic Tissues"

_biomolecules, 2020, doi:10.3390/biom10050766_

Round 1

Reviewer 1 Report

This is an interesting paper from Tabat and colleagues, who utilized ex-vivo experiments of human biopsies to assess the effects of butyrate on acute C48/80-induced colonic hyperpermeability and altered tight junction protein gene expression. The novelty of their work was to bring a more translational approach to challenge previous favorable findings of butyrate on intestinal barrier function from the literature. In general, this is a well-written and straightforward study with clear objectives and research design. The novelty of using human biopsy experiments is a strong point of this study. The authors mentioned that only results from experiments with increased permeability were included in their data. However, some points should be further clarified/discussed by the authors: 1) the age range of the healthy volunteers is broad (18-65 yo). Aging is a key factor affecting intestinal barrier and mast cell function…no clear data is shown how samples from older individuals differ from younger ones. What about gender-related effects…Were age and gender influential factors in mast cell degranulation in these biopsy samples? 2) Oral contraceptives were not an exclusion criteria for the  population sampling…this reviewer has concerns that oral contraceptives may have influenced their findings…mast cells express estrogen receptor-α and estradiol treatment may affect IgE-induced mast cell degranulation…whether oral contraceptives could directly influence intestinal barrier permeability is unclear…but they may have affected mast cell function…would be interesting to have toluidine blue staining of the biopsies to  assess mast cell degranulation. 3) tight junction protein gene expression may not clearly elucidate the tight junction uncoupling of the cell membrane junctional complex…immunofluorescence labeling of the targeted tight junctions would be needed to confirm whether paracellular epithelial barrier function (at the cellular level) wasn’t affected by butyrate. That would complemment the Using chamber data.  

Reviewer 2 Report

The manuscript assesses any potential effects of butyrate on intestinal hyperpermeability induced by a chemical stressor, using human colonic biopsies as the model system. The authors investiagte TER (permeability) and also mRNA expression of tight junction proteins. The novelty of this study is the use of human colonic tissue in which to check for any effect of butyrate- as mnetioned by the authors, the effects of butyrate has primarily been studied to date in mice or in vitro cell models.

Although the results are not terribly exciting, because no effects of butyrate were observed, the results are none the less important. The study highlights the need for further investigation on the effect of metabolites such as butyrate on gut barrier integrity and importantly shows that it is important to consider the models being used, and to incorporate studies in human tissue, so as to get a more complete picture.

Overall. the manuscript is very well written, it reads well, and the data are presented in a clear and concise manner. I have no issues with the science presented and no recommendation for further work.

Author Response

Reviewer 2 had no issues with the science presented and no recommendations for further work.

We thank Reviewer 2 for his/her time and effort to review our manuscript.

Round 2

Reviewer 1 Report

The authors have addressed most of my concerns. No further queries raised.